



# Methods for Estimation of Ionospheric Layer Height Characteristics from Doppler Frequency and Time of Flight Measurements on HF Skywave Signals

Kristina Collins[1], Steve Cerwin[2], Philip J. Erickson[3], Dev Joshi[4], Nathaniel Frissell[4], and Joe Huba[5]

[1]Case Western Reserve University, Glennan Bldg 9A, 10900 Euclid Avenue, Cleveland Ohio 44106, USA
[2]HamSCI, c/o Dr. Frissell, USA
[3]Haystack Observatory, Massachusetts Institute of Technology, 99 Millstone Road, Westford, MA 01886, USA
[4]Department of Physics and Engineering, University of Scranton, Scranton, PA 18510-4642, USA
[5] Syntek Technologies, Inc. 2751 Prosperity Ave. Suite 460, Fairfax, VA 22031, USA

**Correspondence:** Kristina Collins (kd8oxt@case.edu)

**Abstract.**

We demonstrate a methodology to estimate ionospheric virtual layer height characteristics using Doppler measurements from frequency locked time standard stations in conjunction with ionosonde measurements and ray-tracing models. We consider data from three events: the solar eclipse of 21 August 2017, a observations of the dawn terminator on 1 October 2019, and a time-of-flight study conducted 29 January 2020. Observations are consistent with a model in which mode splitting originates from different path length velocities associated with single and multiple hop modes as the virtual layer height changes. Support for this hypothesis comes from the complementary processes of 1) calculating Doppler shifts from virtual layer height changes and virtual layer height changes from Doppler shifts, and 2) the analysis of intermittent low-Doppler shift modes including correlation with ionosonde observations to help identify multihop propagation modes. We find that observations are in good agreement with measured data and simulations. We also find that the use of a precision frequency standard, such as a GPS-disciplined oscillator, at the receiving station is vital for ionospheric height measurements, since small errors in frequency estimation can lead to uncertainties on the order of tens of kilometers in resulting estimations of ionospheric height. The methods discussed herein provide a means to calculate path length estimates from distributed stations when integrated with other ionospheric measurements, helping to address the problem of under-sampling of the bottomside ionosphere.

## 1 Introduction

Ionospheric variability, and its effects on radio wave propagation, have been frontier scientific topics in upper atmospheric physics since the beginning of the radio era more than a century ago. At high frequency (HF, 3-30 MHz), the nature of magnetoionic propagation means that signal paths are heavily refracted by the dispersive, magnetized ionospheric plasma (Budden (2009)). As ionospheric conditions change, time-dependent changes in the phase path length of signals give rise to apparent changes in received frequency, or Doppler shifts.





Doppler shift measurements employing coherent HF frequency beacons, provided by dedicated transmitter networks and/or highly coherent national time standard signals such as provided by the US National Institute of Standards and Technology (Lombardi (2002)), have long been used in remote sensing studies of ionospheric conditions. Beacon based studies have for example derived information on changes in ionospheric height due to traveling ionospheric disturbances (Georges (1968)), and

time standard stations have even been recently used to construct HF imaging radars using large radio telescopes (Helmboldt et al. (2013)). Recent progress in commercially available technology has allowed Doppler measurements at HF to be straightforward and accurately made with coherent radio receivers using local oscillators locked to global navigation satellite system based time distribution, and data can be recorded by inexpensive single-board computers. Due to the low unit cost, the density of Doppler receiver stations can be much higher than the density of transmitting beacons, since receivers generally do not require formal

licenses.

Recently, interest has grown in developing crowd sourced, distributed networks of HF receiver stations for geospace environmental monitoring. These meta-instruments (i.e., networks of small instruments effectively operating as a single large instrument), operated by radio amateurs and shortwave listeners, have the potential to greatly improve the density of instrumentation for bottomside ionospheric sampling. Recent studies by Frissell et al. (2018, 2019); Frissell et al.; Perry et al. (2018)

have demonstrated the scientific validity of qualitative HF observations in these types of networks, using existing amateur radio equipment.

Subsequent processing of these observations for scientific studies of ionospheric conditions and variability requires further analysis to extract information from the direct Doppler observations. This is particularly true given the non-unique nature of the measurements since changes in ionospheric parameters such as peak electron density, layer height, and layer width can at

times produce identical Doppler shifts (Lynn (2009); Chilcote et al. (2015)). Analysis must therefore use reasonable ionospheric models, prior information, or other equivalent techniques in order to stabilize the solution.

In this study, we target ionospheric refraction height information, providing analysis of HF Doppler receiver observations through a quantitative approach relating Doppler shift to ionospheric height. The approach calculates an inferred change in refraction height from observed Doppler data. We also perform correlative analysis of datasets in a combined attempt to

understand specific elements of Doppler shifts and mode splitting observed over a single transmit-receive pair in the continental United States at mid-latitudes.

The study is laid out as follows: In Section 2, we describe the method for tracking the Doppler shift of the carrier from a time standard station to a single receiver, and deducing an inferred layer height. This analysis also employs observations made with vertical incidence ionosondes, specifically the Boulder and Idaho National Laboratory ionosondes maintained through

the Lowell Digisonde network, and uses raytracing simulations based on ionospheric models (2.3). Section 3 applies analysis to three events to which each of these methodologies have been applied on a single refractive path. Section 4 focuses on the phenomena observed in these datasets, including changes of ionospheric layer height, propagation mode splitting and multipath, and the impacts of timing precision on layer height estimation.



## 2 Methodology

### 55  2.1 Remote Sensing Network Description

Data used in this study was collected at separate times on a transmit-to-receive path originating at time standard station WWV in Fort Collins, Colorado (CO) and terminating at either amateur station WA5FRF in Texas or station WA9VNJ (Wisconsin). WWV uses a standard amplitude modulated (AM) waveform to transmit time and frequency information. The stations are mapped in Figure 1, along with path of totality data for the 2017 solar eclipse.

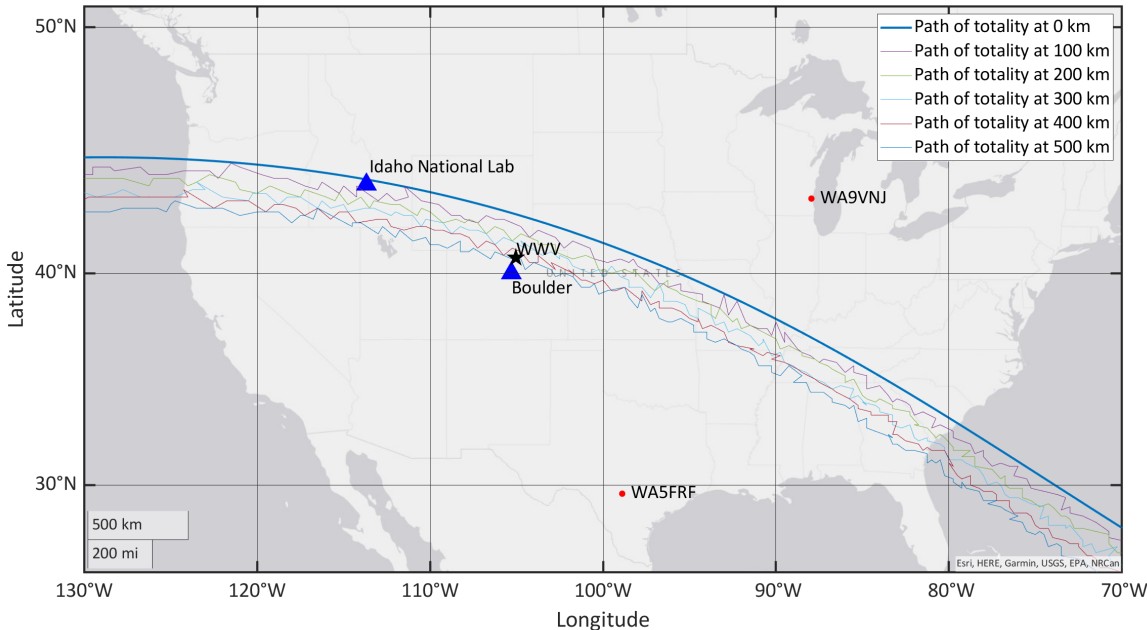

**Figure 1.** Map showing locations of stations, 2017 paths of totality at elevation. The primary beacon used in all Doppler data herein is WWV near Fort Collins, CO; WWVH, in Kauai, HI, is not shown. Amateur radio stations WA9VNJ and WA5FRF are marked, as are the ionosondes used for comparison.

Multiple ionospheric datasets including Doppler frequency spectrograms, ionosonde data, time of flight (TOF), and supporting raytrace simulations were acquired on different days, frequencies and events. These are summarized in Table 1.

For the WA5FRF receiver data, analysis also used separate layer height information provided by the ionosonde with URSI code BC840, located in Boulder, CO, and operated by the United States Air Force. Data used is available through the Digital Ionogram Database (Reinisch and Galkin, 2011). We specifically use the hmF2 parameter, which is the peak height of the F2 65  layer as calculated from ionogram measurements. For the WA9VNJ dataset, ionosonde data from ionosonde IF843 at Idaho National Laboratory was used. The locations of these ionosondes are also shown in Figure 1.



| Event and Phenomena of Interest | Path | Freq. | Dataset | Raytracing Simulation | | Obs. | Results |
|---|---|---|---|---|---|---|---|
| | | | | Model | Citation | | |
| August 2017 Solar Eclipse: Solar eclipse obscuration | WWV ⟶ WA9VNJ | 10 MHz | Reyer (2017) | SAMI3 | Collins (2021c) | Sect. 3.1 | Figs. 5, 6 |
| October 2019 Festival of Frequency Measurement: Morning transition | WWV ⟶ WA5FRF | 5 MHz | Cerwin (2019c) | IRI | Collins (2021a) | Sect. 3.2 | Figs. 7, 9, 10 |
| January 2020 Timing Study: Multipath propagation | WWV ⟶ WA5FRF | 5 MHz | Cerwin (2020) | IRI | Collins (2021b) | Sect. 3.3 | Figs. 12, 14, 15 |

**Table 1.** An overview of the datasets and raytracing simulation results used in this paper, with the sections in which they are referenced.

## 2.2 HF Doppler Methodology

Under the Martyn equivalent height approximation (Martyn (1935)), Doppler shift from the ionosphere will occur when the effective ionospheric layer height changes, since under this scenario the received frequency will deviate from a known transmit frequency as an electromagnetic wave is reflected from the equivalent of an approaching or receding mirror point. Such apparent motion occurs frequently during certain types of solar, ionospheric and geomagnetic events when radio waves are refracted by a time-varying ionosphere. Hence, Doppler shifts can be analyzed as signatures to understand aspects of ionospheric dynamics due to a variety of disturbance time phenomena. Previous beacon based studies have used this approach to derive ionospheric information (Davies et al. (1962)), and such work cites advantageous qualities of this approach such as (1) relative simplicity and low cost of operation, (2) ease of data storage and convenience of data presentation, and (3) continuous monitoring of the ionosphere.

Various techniques have been described in the literature to interpret Doppler shifts of a continuous wave (CW) carrier induced due to an ionosphere in motion. Fourier analysis of a signal reflected from the ionosphere can be used to decompose the various Doppler shifted components of the transmitted signal. The composition of the Fourier components can be helpful in understanding the ionospheric state when the reflection occurred. In particular, a geomagnetically disturbed ionosphere with distributed density structure would result in a spread in received Doppler spectrum, whereas a nominal undisturbed ionosphere would result in more sharply defined Fourier components as reflections occur from the more uniform motion in (potentially multiple) stratified ionospheric layers. Other analysis techniques include forward modeling the ionosphere as a series of time-





varying parabolic layers, parameterized by peak height, maximum electron density/critical frequency, and parabolic thickness,
as per Lynn (2009).

In general, given a particular ionospheric configuration, it is not straightforward to obtain an exact expression for the effective Doppler frequency shift, as multiple reasons can cause refractive index changes and therefore phase path length changes. For instance, refractive index change during solar flares can occur due to an increase in ionization rate and hence ionospheric electron content. In other cases where significant ionospheric electron density increase does not occur, Jacobs and Watanabe
(1966) enumerate possibilities that cause change in phase path length, including change in intensity of the geomagnetic field and change in electron density spatial distribution.

In this study, we use the equivalent height model to analyze for an effective ionospheric layer height change. While earlier work focused primarily on received Doppler shift of the AM carrier signal (Collins et al. (2022)), we also examine further information contained in the modulated signal, specifically short bursts of 5 cycle 1000 Hz audio "ticks" that occur precisely
once per second as regulated by atomic clocks at the transmitter.

### 2.2.1  Calculation of Layer Height Changes from Measured Doppler Data

The Doppler frequency shift as a signal travels from transmitter to receiver is given by :

$$\Delta f = -(f/c) \cdot dP/dt \tag{1}$$

where $P$ is the phase path of a ray for frequency $f$ (the transmitter frequency) and $c$ is the free-space velocity of light.

From Equation 1, Doppler shift responds to the time rate of change in path length, $P$. (The polarity of Doppler shift is such that a decreasing path length causes a positive shift in frequency and an increasing path length causes a negative shift.) Therefore, if Doppler shifts are measured, the path velocity can be calculated and the actual change in path length calculated by integration. Once path length change is determined, the change in effective layer height can be calculated through the use of a geometric model such as the Martyn equivalent virtual height model shown in Figure 2. We use this forward model to
relate ground distance, layer height and launch angle during analyses. Because the integral is definite and the initial condition unknown, an estimate for the constant of integration must be obtained from another source. This can be obtained from ionosonde derived effective ionospheric electron density profiles if one is available near the apogee of the propagation path, assuming that significant refraction occurs largely near that point. Combined with ray tracing and empirical ionospheric models, the external information is used twice: first to fix the operating point on the nonlinear height vs. path length curve set by path geometry, and
again to fix the starting point for calculated height changes.

### 2.2.2  Height Estimation Algorithm

Ionospheric height is estimated by the following procedure, shown in Figure 3.

1. Digitize Doppler spectral data using a suitable time increment between data points.




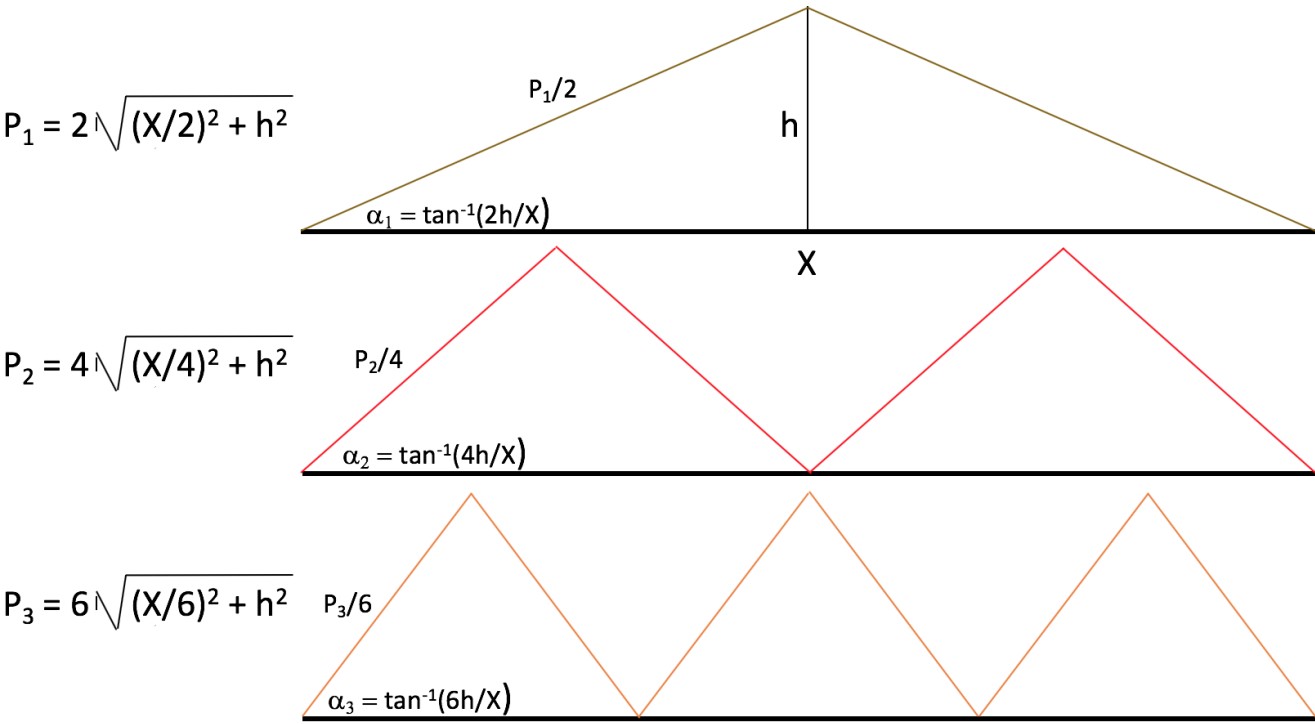

**Figure 2.** The geometric Martyn equivalent height relationship between stations and control points (i.e., ionospheric virtual reflection points) causes Doppler shift from multihop paths to reflect change in ionospheric height. This geometric model is required to calculate relative path lengths (and subsequent times of flight at the speed of light) for 1, 2, and 3 hop modes that reflect from a common virtual height over a fixed ground distance. It is this relationship that supports the mode splitting hypothesis.

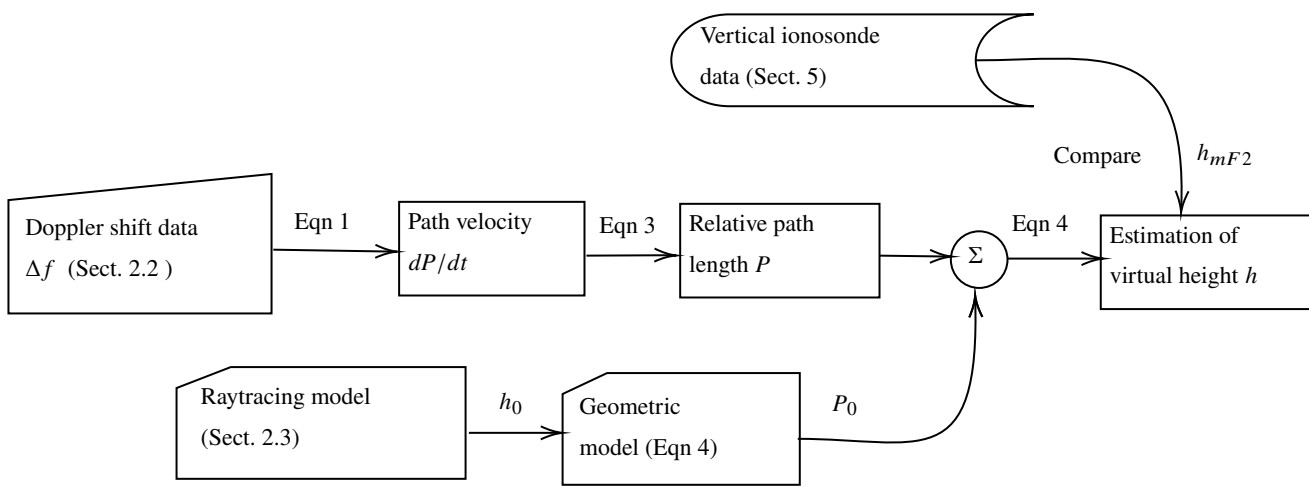

**Figure 3.** Flowchart illustrating the procedure in Section 2.2.2.





2. Convert Doppler frequency to equivalent path velocity. The conversion factor is frequency dependent and equals the ratio of $-c/f$, per Equation 1. (For example, to convert the Doppler shift in Hz to path velocity in m/s at 5 MHz, the Doppler shift would be multiplied by -59.9585; at 10 MHz, it would be multiplied by -29.9792.)

3. Calculate path length increment $D_P$ for each time increment $k$ in the record

$$D_{Pk} = D_{vk} \cdot D_t, \tag{2}$$

where $D_{Pk}$ represents the kth incremental change in path length, $D_{vk}$ represents the kth path velocity increment, and $D_t$ represents the time increment. The average of the start and end velocities for each increment was used to improve accuracy.

4. Integrate path length increments to compute relative path length vs. time, discretized in $M$ increments.

$$P = \Sigma_{N=1}^{M} D_{PN} \tag{3}$$

5. Select an initial reference layer height $h_0$ from raytrace simulations or actual ionosonde measurement. (The relationship between layer height and path length is nonlinear and therefore the rate of change of path length with height depends on actual layer height.)

6. Calculate the initial absolute path length $P_0$ from $h_0$ using the geometric model as applied to the particular mode analyzed. For a number of hops $N$ across a distance $X$ from the transmitter to the receiver, the path length for a height of reflection $h$ may be calculated:

$$P_N = 2N \sqrt{\frac{X}{2N} + h^2} \tag{4}$$

This relationship between path length and height is illustrated in Figure 2 for 1, 2, and 3 hop modes.

7. Calculate absolute path length vs. time by adding $P_0$ to the relative path length calculated using Equation 3 above. This is illustrated by the summing junction in Figure 3.

8. Calculate layer height from absolute path length for the appropriate propagation mode, using Equation 4, rearranged to solve for $h$.

## 2.3 HF Raytracing

Since HF signal travel through the ionosphere is refractive, background ionospheric electron density information is needed as well as a geometric optics based raytracer to determine a particular signal path at a given time and location. Raytracing simulations were prepared using the PHaRLAP toolkit (Cervera and Harris (2014)) and two ionospheric models: the empirical International Reference Ionosphere 2016 (IRI) (Bilitza et al. (2017)) and an eclipsed version of the first-principles physics-based



SAMI3 model by Huba and Drob (2017). In the latter case, the eclipse was simulated by running SAMI3 with a solar extreme UV radiation mask with a maximum obscuration of 0.85 applied to model inputs. The simulated cases are listed in Table 1.

Although these models do not incorporate electron density variability on short spatial or temporal scales (aside from the eclipse applied to the special run of SAMI3), they can provide phenomenological insight into the causes of features observed during Doppler studies. These models are also used, as described in Section 2.2.2, to provide a starting estimate for ionospheric height at the apogee of a single hop path.

In each case, a two-dimensional raytrace was simulated from WWV in Fort Collins to the receiving station where the observations were conducted. Azimuth was found via the great circle path from WWV to the receiving station. The takeoff angles for rays reaching the receiving station were determined by an iterative algorithm. Only single-hop paths were included in the simulation.

An additional expository multihop simulation (Figure 4) was also performed using a 2D raytrace in IRI 2007, for the January 2020 dataset. These results were not used quantitatively, but do provide a comparative and notional illustration of multihop modes.

## 3 Case Studies

Three data collections were analyzed in this study, as listed in Table 1.

### 3.1 August 2017 Solar Eclipse

The August 21, 2017 total solar eclipse across the continental United States (CONUS) captured public attention in the United States for its visual display, but also provided an opportunity for ionospheric study and radio science. Through creation of a temporary reduction in extreme ultraviolet (EUV) radiation into the upper atmosphere, the 2017 eclipse caused reductions in ionization and hence in electron density. Radio propagation was subsequently affected on medium- and high-frequency bands (MF and HF, 300 kHz to 30 MHz).

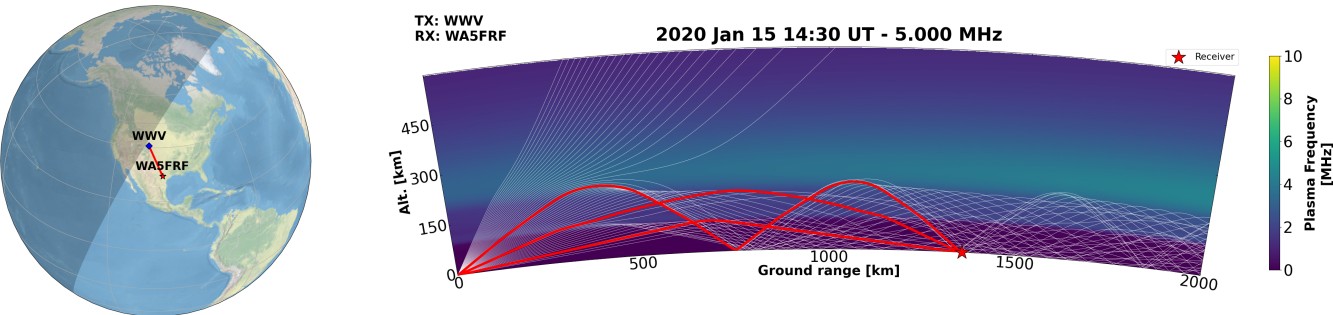

**Figure 4.** A 2D raytrace simulation that highlights simultaneous multihop propagation modes from WWV to WA5FRF. Made with Natural Earth (Met Office (2010 - 2015)).




During the 2017 eclipse, amateur radio operator Steven Reyer WA9VNJ recorded the apparent carrier frequency of the NIST radio station WWV on 10 MHz in north suburban Milwaukee, Wisconsin (Reyer (2017)). Observations were simulated using the PHaRLAP raytracing toolkit and the eclipsed SAMI3 ionospheric model (Collins (2021c)).

A plot of the observed Doppler shift is shown in Figure 5. The plot shows conditions at the geographic midpoint between WA9VNJ and WWV, as calculated by the haversine method, which is taken to be the approximate reflection point: the start and end of the eclipse at that point are shown, as is the obscuration at 300 km elevation. Compared to control data from the previous day, the recording from the day of the eclipse shows pronounced variation during the eclipse itself. The elevated Doppler shifts on the order of 1.3 Hz between 1400 and 1600 UTC (locally, late morning) transition into negative Doppler shifts within an

envelope of -0.2 to -0.4 Hz as the eclipse obscuration increases. This is consistent with expectations of a lowering virtual height (and hence a reduced path length, and negative Doppler shift) with a decrease in photoionization. As the obscuration reduces, the trend in the Doppler shift is reversed: it becomes positive within a positive envelope of 0.2 to 0.4 Hz. A manual envelope superimposed over this trend, indicated by the dotted line, ties this figure to the vertical ionosonde results from the Idaho National Lab ionosonde in Figure 6. Similarly, this plot shows the results from 20 and 21 August 2017, with elevated

obscuration (this time, directly above the vertical ionosonde) mapped on the plot. Here, the envelope from the previous figure is integrated to represent estimated shift in hmF2 and superimposed on the plot: the estimated magnitude of the shift in layer height is shown to be in rough qualitative agreement with the ionosonde measurement in the path of totality.

### 3.2  October 2019 Festival of Frequency Measurement

Amateur radio operator Steve Cerwin WA5FRF recorded frequency data (Cerwin (2019a)) in the 5 MHz time standard signal of

WWV during the WWV Centennial Festival of Frequency Measurement, described by Collins et al. (2022). In this crowdsourced campaign, amateur radio operators used a standardized approach to estimate Doppler shift from the carrier of WWV. This paper uses the data collected with the standardized approach outlined in Cerwin (2019b), as well as observations made from spectrum recordings (Cerwin (2019c)).

Figure 7 shows Doppler shift data collected during the morning transition on October 1, 2019. On the left is the spectrogram

acquired by Spectrum Lab using a receiver that did not have a GPSDO but was calibrated using a daytime 5 MHz WWV skywave signal. On the right is the data after manual digitization.

Figures 8 and 9 show the changes in path velocity and length calculated from this data according to the procedure above.

### 3.3  January 2020 Time of Flight Study

Using not only the constant AM carrier but audio modulation information, a dedicated time of flight study of WWV signal

propagation was undertaken at station WA5FRF on 29 January 2020 (Cerwin (2020)).

In this study, time standard signals were observed, and in particular precise 1000 Hz tone burst signals or "ticks" (5 cycles long) were demodulated from the WWV AM signal using an Icom IC-7610 commercial HF receiver and manually recorded on an Agilent DSO5034A digital oscilloscope. Relative time synchronization at the receiving station was provided by the rising edge of a 1 pulse per second (pps) output from a GNSS/GPS local oscillator that was globally coherent with the GPS satellite





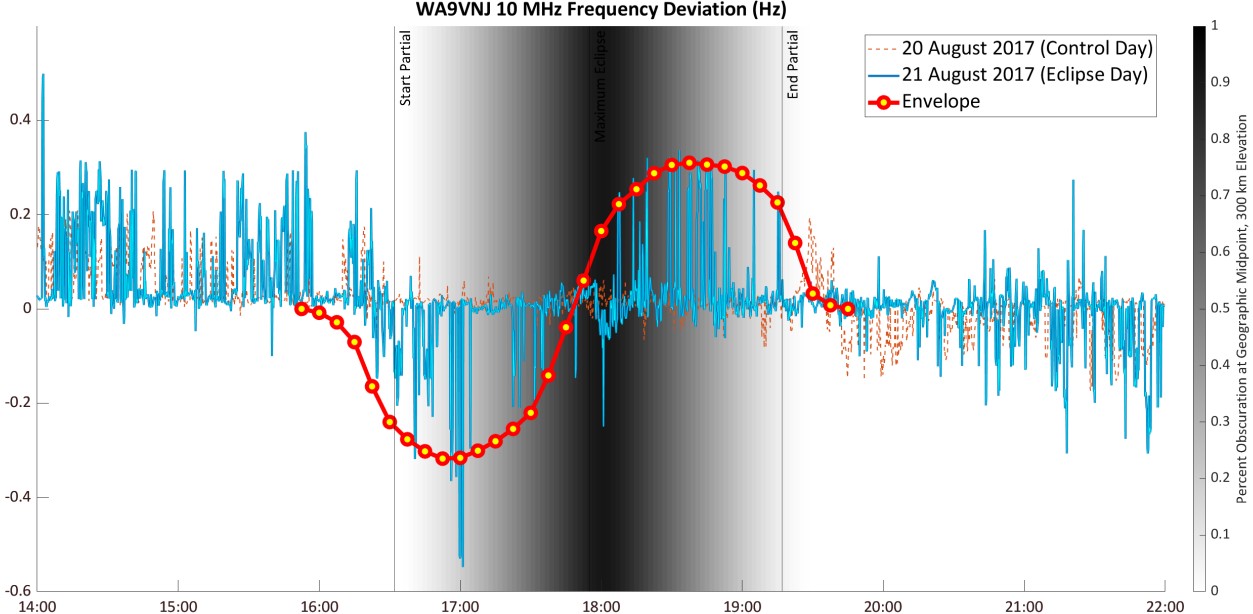

**Figure 5.** Plot of data collected by WA9VNJ, as described in Section 3.1 and reported by Reyer (2017). The background shading represents the eclipse obscuration at the geographic midpoint between WWV and WA9VNJ (42.267, -96.667). A manually determined envelope is superimposed over the data.

constellation clocks. The jitter and accuracy associated with the rising edge of the GPSDO 1 pps output is generally well within 100 ns, per Lombardi et al. (2005). Since the desired measurement precision is on the order of 0.01 ms (10 us or 10,000 ns) the uncertainty in the 1 pps output is negligible. The throughput delay from the RF input to the audio output of the Icom IC-7610 was measured at 4.0 ms, which matches the manufacturer's specification, by feeding an AM modulated burst similar to the WWV timing tick into the receiver RF input and measuring the delay to the corresponding receiver audio output. A block diagram of the setup is shown in the upper half of Figure 11. Doppler shift information was also collected during the observation using a setup depicted in the lower half of Figure 11.

For the January 2020 time of flight study (cf. Sect. 3.3), timing data collected over a 4 hour window was plotted on a scatter diagram, shown in Figure 12. Results showed that times of flight were statistically clustered in groups. These clusters correlated well in a general sense with those calculated from a geometric model of the time delays associated with single and multiple hop modes over the propagation path from Colorado to Texas, as shown in Figure 13.

Note that while both WWV and WWVH were being received at the same time during this experiment, this proved not to be an issue for separating timing ticks from these stations due to the large difference in propagation time between WWV and WWVH at WA5FRF (≥ 17 msec).





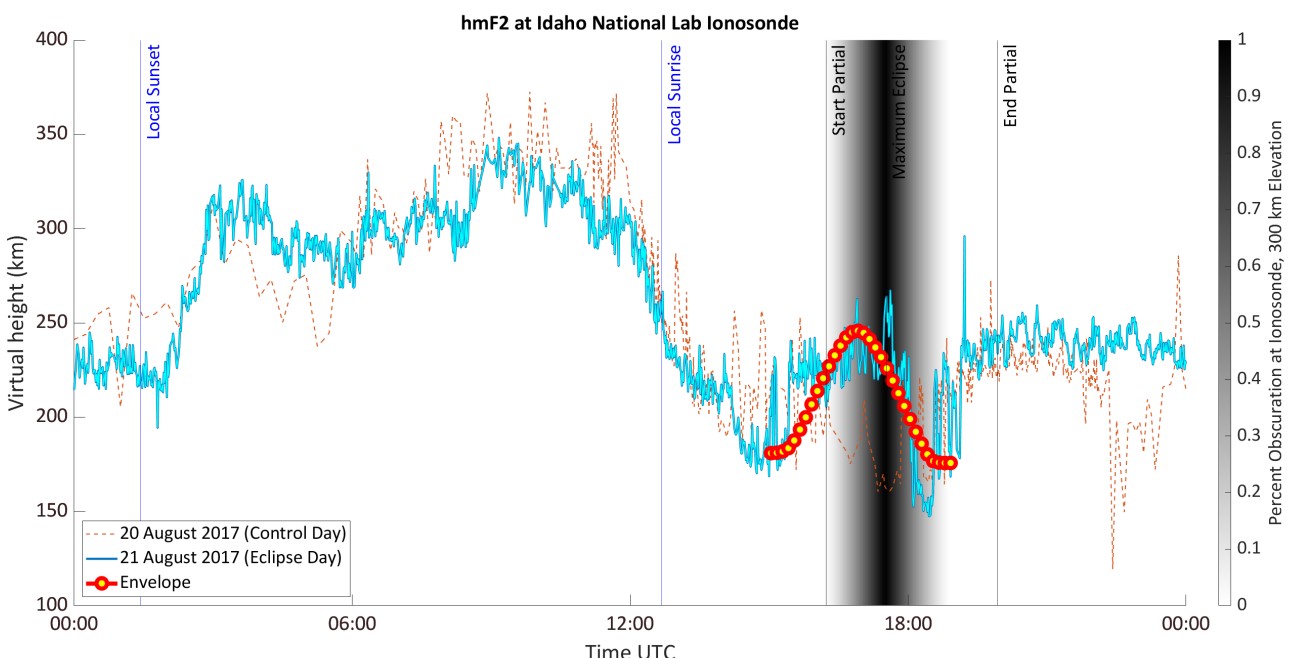

**Figure 6.** Results from Idaho National Lab ionosonde on the day of, and prior to, the 2017 eclipse. The envelope marks the expected virtual height shift corresponding to the envelope marked in Figure 5. Vertical lines indicate the sunrise, sunset and eclipse times from the viewpoint of an observer on the ground.

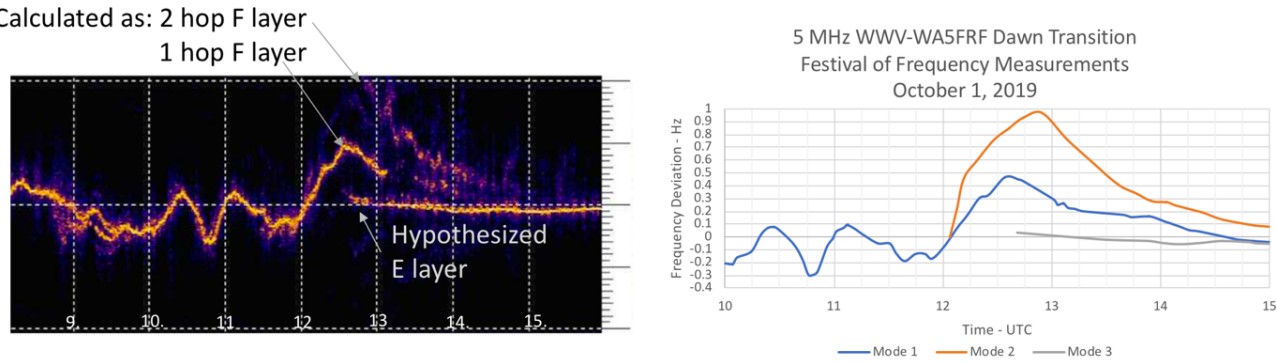

**Figure 7.** Left: Spectrogram of 5 MHz WWV during morning transition on 1 October 2019, collected by WA5FRF. The vertical axis is ±1 Hz. Right: Spectral data, digitized and recorded in Excel spreadsheet. The carrier is seen to fragment, indicating multiple simultaneous propagation modes.



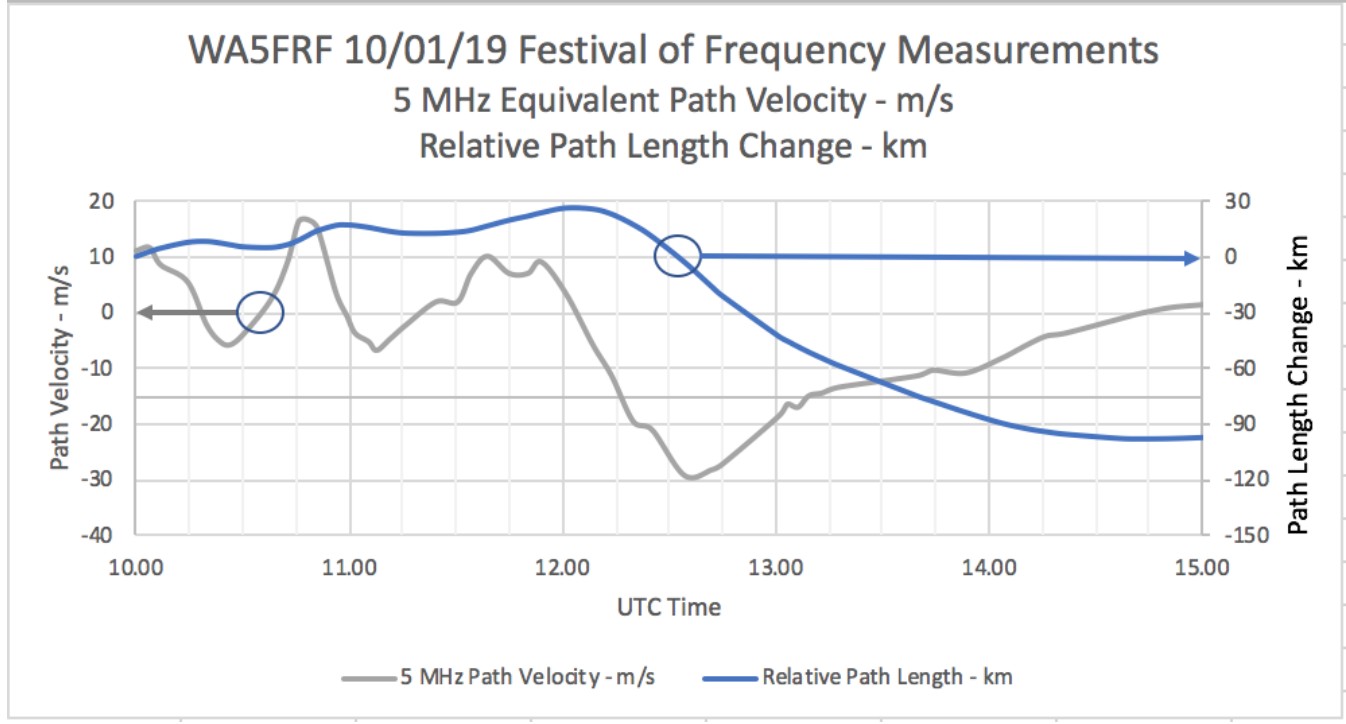

**Figure 8.** Equivalent path velocity (gray line, left axis) calculated from Doppler shift and associated relative change in path length (blue line, right axis), calculated per Equation 3.

## 4 Discussion

### 4.1 Time of Flight Considerations in HF Propagation

The pulse width used in the WWV transmission is greater than the length of the observed time delays between multiple modes, causing the primary and delayed pulses to overlap, as shown in Figure 14. To model this effect, the top row of plots shows SPICE simulations for superposition delays of 1, 2, and 3 ms. The net effect of overlapping arrivals is to lengthen the overall pulse. Beneath each simulation is plotted an oscilloscope recording showing actual WWV on-the-air captures for the corresponding simulation condition. Note the first cycle of the primary mode and the last cycle of the delayed exhibit no overlap and are free from superposition distortion.

A preferred measurement technique would measure TOF of the primary pulse by referencing the leading edge, and delayed arrivals by referencing the trailing edge. For this technique, the rising edge of a GPSDO's 1 pps output would be used as a reference mark. Time measurement would need to take into account the time delay through the receiver by using repeatable on-time markers on the pulses. However, for the experiments reported here, the actual beginning of the pulse was at times difficult to discern under noisy conditions because the exact starting point is near zero amplitude and can be masked by noise. Therefore, the middle of the first positive half-cycle of the timing tick was used as the on-time marker and the corresponding



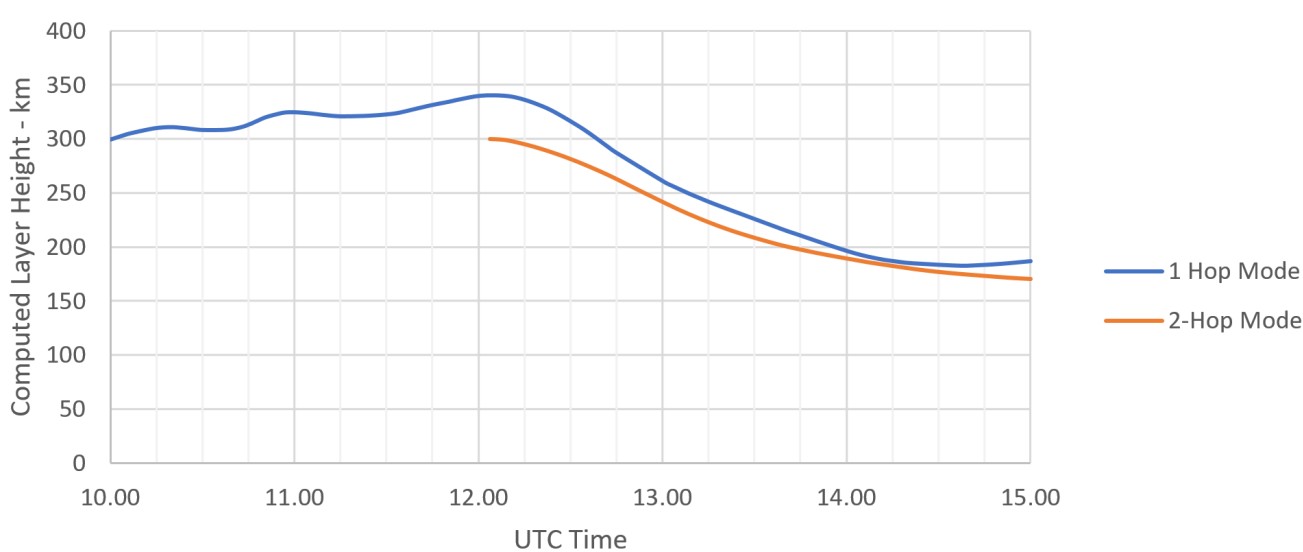

**Figure 9.** Layer heights calculated from multimode Doppler data. The top trace was calculated from the Doppler track labeled "1-hop F layer" and "Mode 1" in Figure 8 using the 1- hop formula from Figure 2. The lower trace was calculated from the Doppler track labeled "2-hop F layer" and "Mode 2" using the 2-hop formula. Both calculations gave very nearly the same layer height profile, lending support to the multi-hop theory for mode splitting.

time difference between this position and actual pulse start position was taken into account. This point on the waveform is at maximum amplitude instead of minimum and is well above the noise floor and much easier to discern.

**4.2 Ionization Layer Height Impacts on HF Doppler Shift**

To examine multiple mode propagation, the following discussion further describes a 5 MHz skywave path from WWV in Colorado to WA5FRF in Texas during a morning transition in January 2020 (Section 3.3).

The raytrace simulation in Figure 4 demonstrates that multiple simultaneous modes are possible during the morning transition between night and day for the time of year shown. The analyses presented here will focus on only one phenomenon: simultaneous 230 one and two hop modes from the F layer during a morning transition. Not shown are other possible modes that include internal reflections between the E and F layers and modes that spend an extended time within a layer (Pedersen rays).





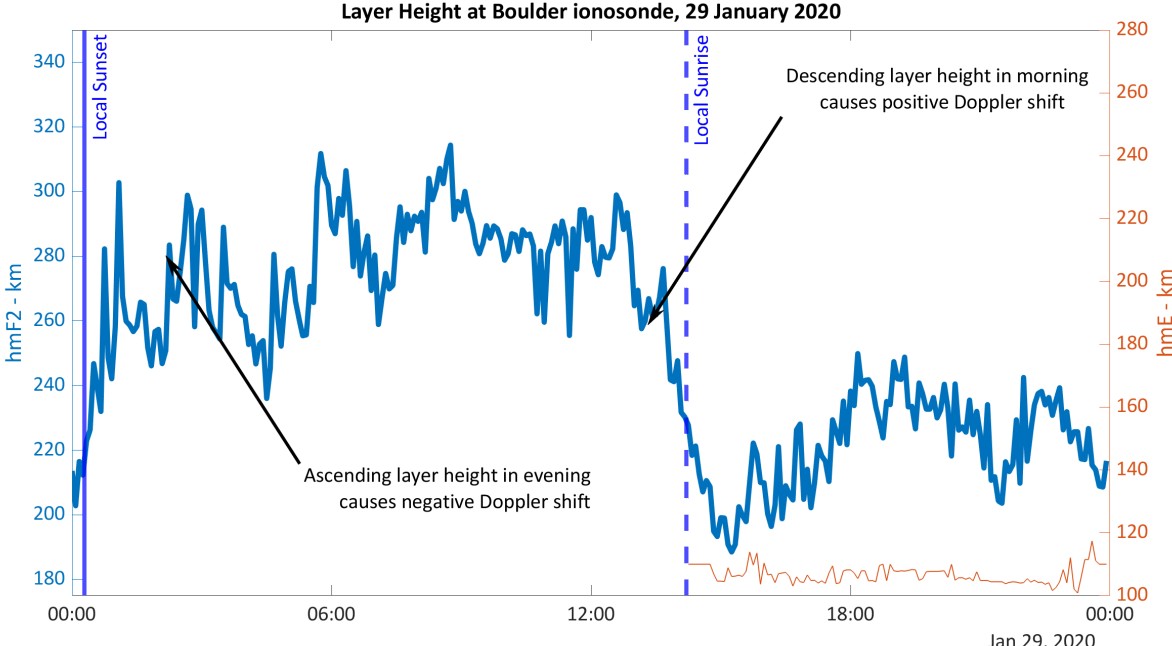

**Figure 10.** A daily plot of layer height (hmE and hmF2) recorded by the Boulder ionosonde operated by UMass Lowell on 29 January 2020.The scaling on both axes is the same, demonstrating that the change in hmE is far smaller than the change in hmF2. A spurious peak in the hmF2 estimation data, caused by poor algorithmic fitting, was removed at 15:40.

Modeling conditions during the event requires an independent estimate of electron density layer altitude. For the event in question, F2 electron density peak layer altitude varied from ~300 km at night to ~200 km in daytime, per observations of F2 peak variations over 24 hours (hmF2) from the Boulder, CO UMass Lowell ionosonde, as shown in Figure 10.

Figure 15 shows frequency spectra of 5 MHz WWV signals transmitted from Ft. Collins, CO and received at amateur radio station WA5FRF site near San Antonio, TX, as described in Section 3.3. The top row of spectrograms covers a 4-hour time span during the morning transition. The bottom row shows spectrograms during the evening transition.

Since Doppler shift responds to the velocity of path length change, and path length between two geographically fixed locations is set by refraction layer height and propagation mode under the equivalent height assumption, maximum Doppler shift occurs 240 where layer height has the fastest time rate of change. In Figure 10, this occurs on the steep slopes where day transitions to night (between approximately 0100z and 0300z) and where night transitions to day (between about 1200z and 1300z). These time periods correspond generally to the negative Doppler swings shown in the lower part of Figure 15 and the positive swings in the upper half.



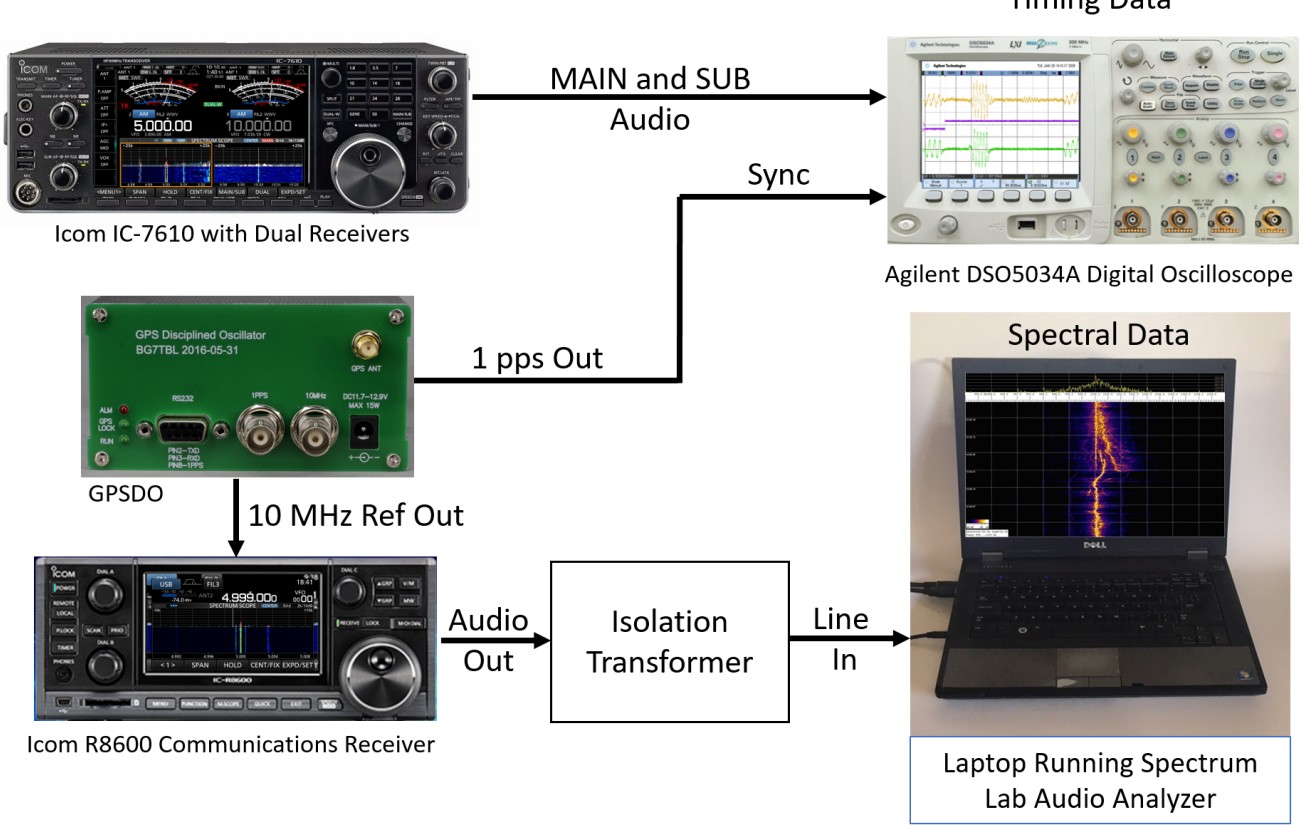

**Figure 11.** Instrumentation used in the timing study described in Section 3.3. The IC-7610 receiver was in AM mode during the timing study (as opposed to upper sideband mode, which is used in Doppler collection.) It was necessary to measure and account for the lengthy propagation delay through the receiver's IF chain. Icom images used with permission.

### 4.3 HF Propagation Mode Splitting

Figure 9 plots the results of the layer analysis approach described in Section 2.2.2. The upper trace is data calculated from the Mode 1 trace, assumed to be from a 1-hop mode in the spectra of Figure 7 using Equation 4. The lower trace is calculated from the Mode 2 trace, assumed to be from a 2-hop mode using the 2-hop formula. Results show that the two traces give comparable values for layer height when calculated from different Doppler tracks, supportive of the multihop hypothesis for mode splitting (cf. Figure 4).

Under this latter hypothesis, one possible cause for mode splitting during the morning transition is the existence of a common refraction layer, widely distributed spatially and descending on simultaneous multiple hop modes. Total path length from transmitter to receiver is a function of the number of hops: multiple hop modes have a longer path length due to the increased




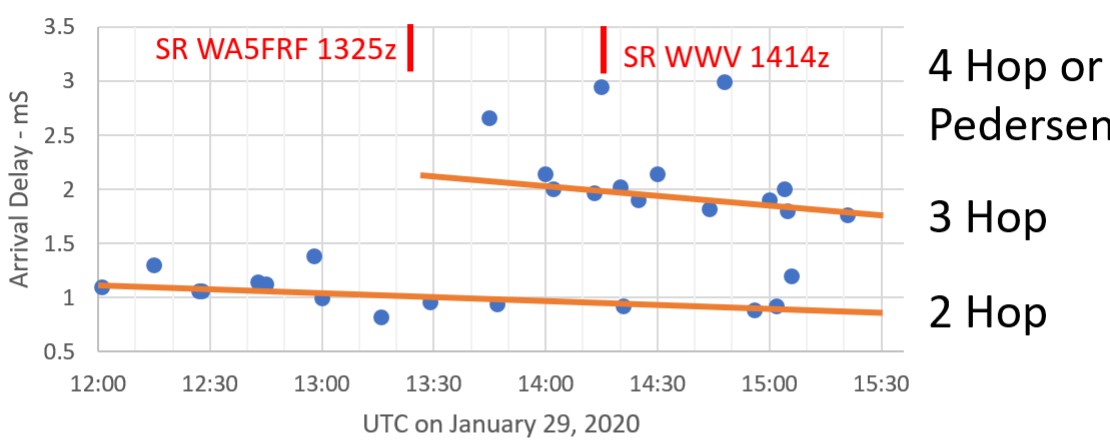

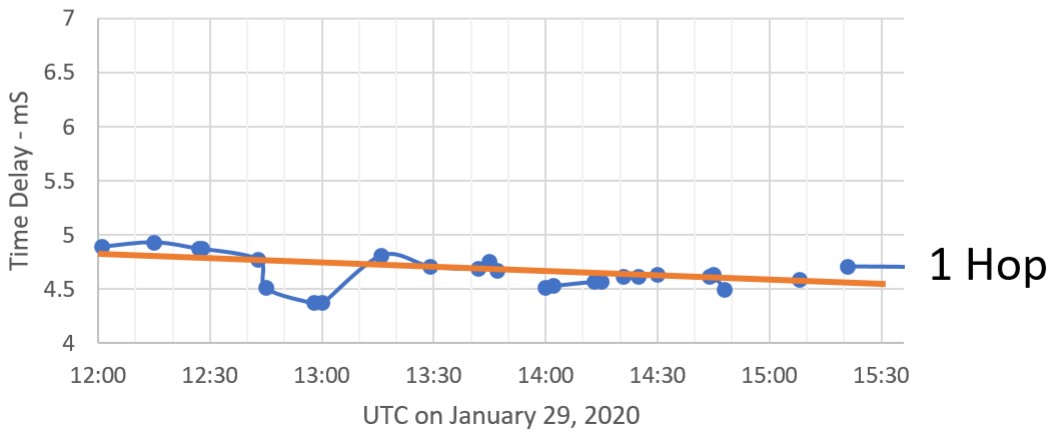

**Figure 12.** Plots from Cerwin (2020) showing statistical clustering in time of flight estimations, indicative of multihop propagation paths. Clustering of arrival times in an overtone progression is consistent with multiple hop modes. Negative slopes are consistent with decreasing path lengths from descending ionization layer. The arrival times of the second and third delayed pulses can be estimated by adding the delay times in the top graph to the Primary arrival times in the bottom graph. The effective virtual height descent rate is available from the plot slopes.



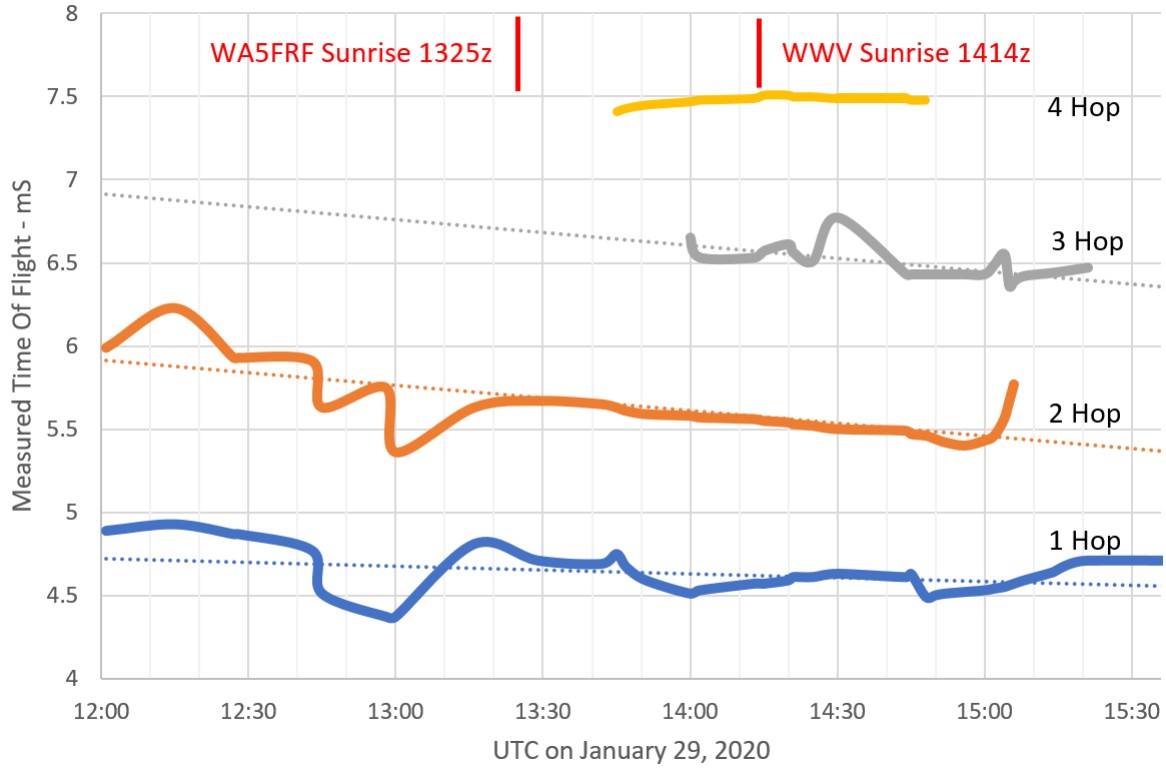

**Figure 13.** Interpolated time of flight data (solid lines) plotted over geometric estimation (dotted lines) Cerwin (2020).

up and down trajectories over the same ground distance. As the common layer height descends, the longer paths containing multiple hops undergo a greater reduction in distance. But all modes shorten over the same time period. More change in distance

during the same amount of time results in a faster velocity of path change for the longer modes. Since Doppler scales with path velocity, the higher order modes exhibit more Doppler shift according to number of hops.

These effects predict the appearance of divergent frequency tracks, or mode splitting, in observed spectra. These were observed in the event reported here, e.g. in the upper half of Figure 15. In particular, a 1-hop mode from the F layer exhibited the lowest Doppler swing and modes with more hops produced correspondingly higher order frequency swings.

During sunrise there can be one to as many as three additional higher order modes that show a positive Doppler shift. Some modes are not continuous but manifest and extinguish abruptly during the course of the transition. This phenomenon is believed to occur because manifestation of a higher order mode is dependent on sufficient ionization to support the higher propagation angle required for the mode. If the ionization is insufficient for required incidence angle, the mode simply escapes into space. As the morning transition progresses, the ionosphere is illuminated at the correct angles for a high order mode but

propagation of the mode does not actually begin until ionization increases sufficiently to return the mode to earth, resulting in



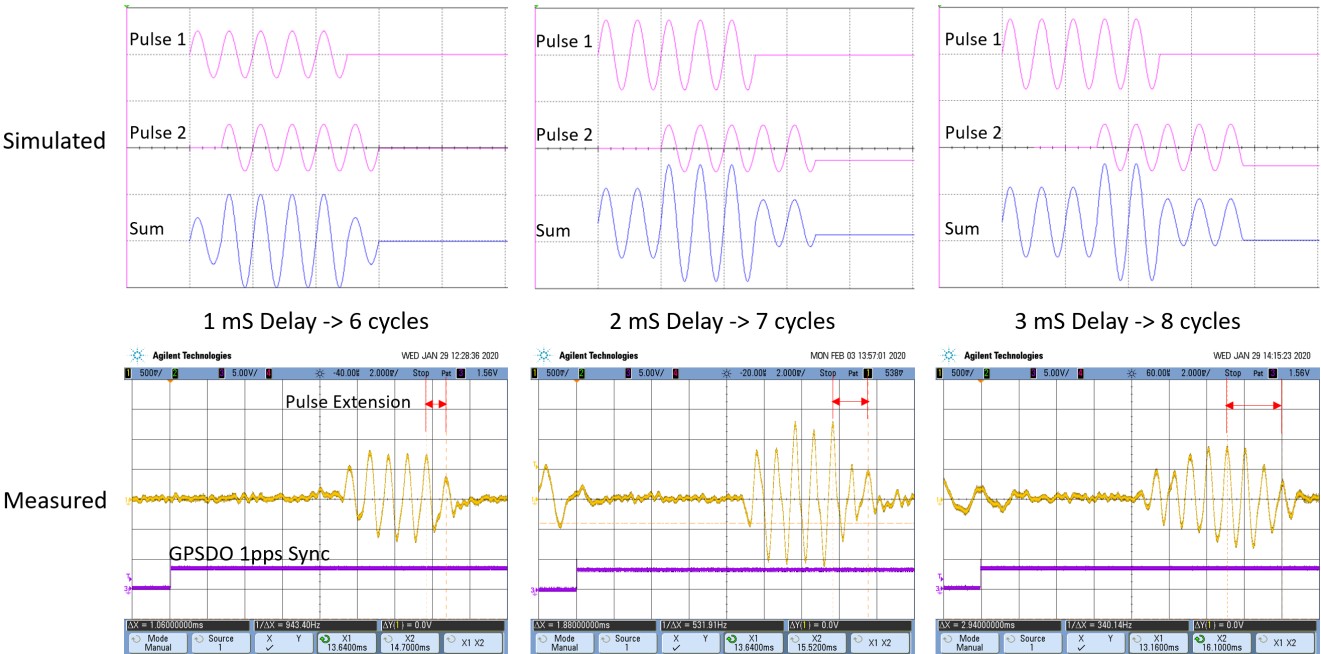

**Figure 14.** Top: Simulated traces in Multisim. Two bursts consisting of 5 cycles of a 1 kHz sine wave representing the WWV timing tick were generated with a variable delay between them. These are shown in the top two traces labeled Pulse 1 and Pulse 2. Then these two waveforms were summed with time delays of 1 ms, 2 ms, and 3 ms to produce the Sum outputs shown in the bottom traces labeled Sum. The time delays between multiple modes are less than the pulse width; therefore, the primary and delayed pulses overlap, lengthening the pulse by superposition.

abrupt manifestation of the mode partway through the transition. The data in Figure 15 was taken during the very low part of the sunspot cycle between Cycles 24 and 25. This effect is present in the datasets described in Sections 3.2 and 3.3.

Note that modes that show little or no Doppler shift are believed to come from the E layer, based on the following observations:

1. **Smaller height changes:** the E layer shows comparatively small diurnal height changes. Ionosonde hmE data shows only about a 20 km change out of 100 km in E layer height. This is in stark contrast with a 100 km change out of 300 km for hmF2 (cf. Figure 10).

2. **Geometry:** a smaller apex height for E layer refraction means there is a smaller change in path length for a given change in height. Therefore, small changes in height result in even smaller changes in path length, and accordingly small predicted Doppler shifts.

3. **Timing:** very steady modes with little Doppler shift occur predominantly in the daytime on 5 MHz. This is consistent with the E layer propagation diminishing at night and dominating during the day for this frequency.





## Positive Frequency Excursions During Sunrise

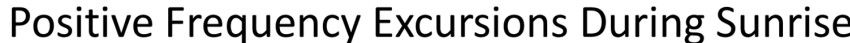

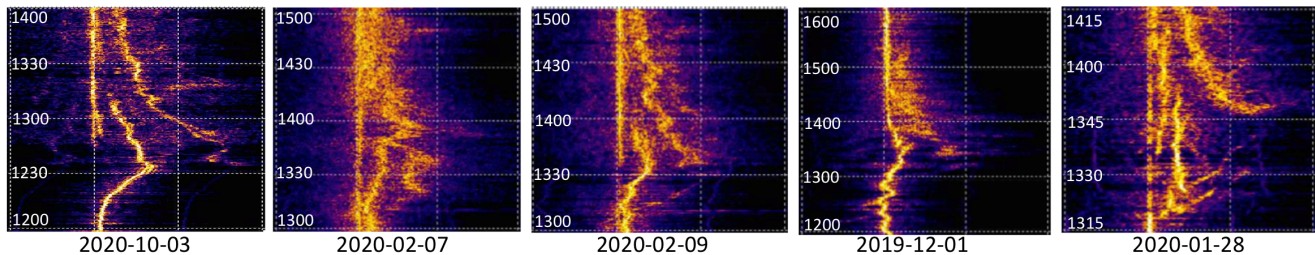

## Negative Frequency Excursions During Sundown

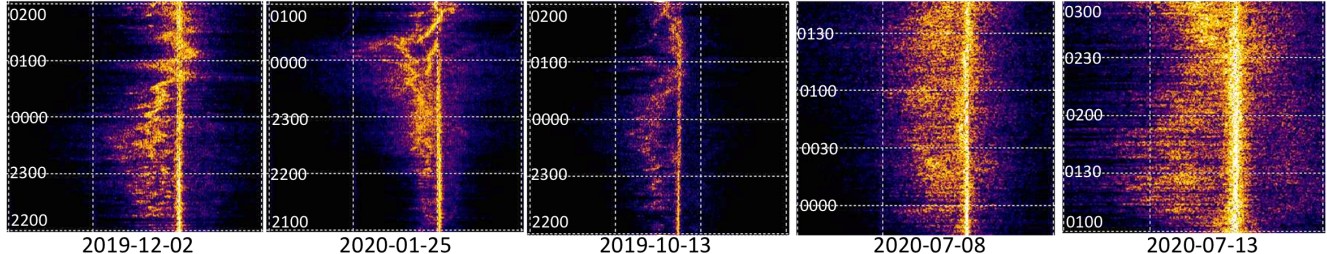

**Figure 15.** WWV spectra received at WA5FRF on 5 MHz during sunrise and sunset showing Doppler shifts, mode splitting, and abrupt mode manifestation and extinguishment.

### 4.4 Impacts of Timing Precision on Height Estimation

A frequency error represents a constant path velocity offset that adds up to a large layer height error when integrated over the time window of the experiment.

The dataset shown in Figure 16 demonstrates the large errors that could occur with as little as 0.01 Hz frequency error at 5 MHz. As shown, a 0.04 Hz range in frequency uncertainty results in a layer height spread of 80 km over a 5 hour time span for this path.

Therefore, this analysis demonstrates a key engineering requirement for distributed Doppler collection: the atomic clock precision of the WWV transmitter and similar precision afforded by a GPSDO at the receiver are requirements for accurate Doppler analyses.

### 5 Conclusions and Future Work

Doppler shifts over a given propagation path have been shown to follow the time derivative of changes in path length, and in turn to act as a function of assumed / effective layer height. The relationship between single and multiple hop path lengths and apogee height can be approximated from a geometric virtual height model. Time of Flight measurements gave results consistent with geometric model and raytrace simulation predictions. Doppler shifts showing mode splitting can be predicted by





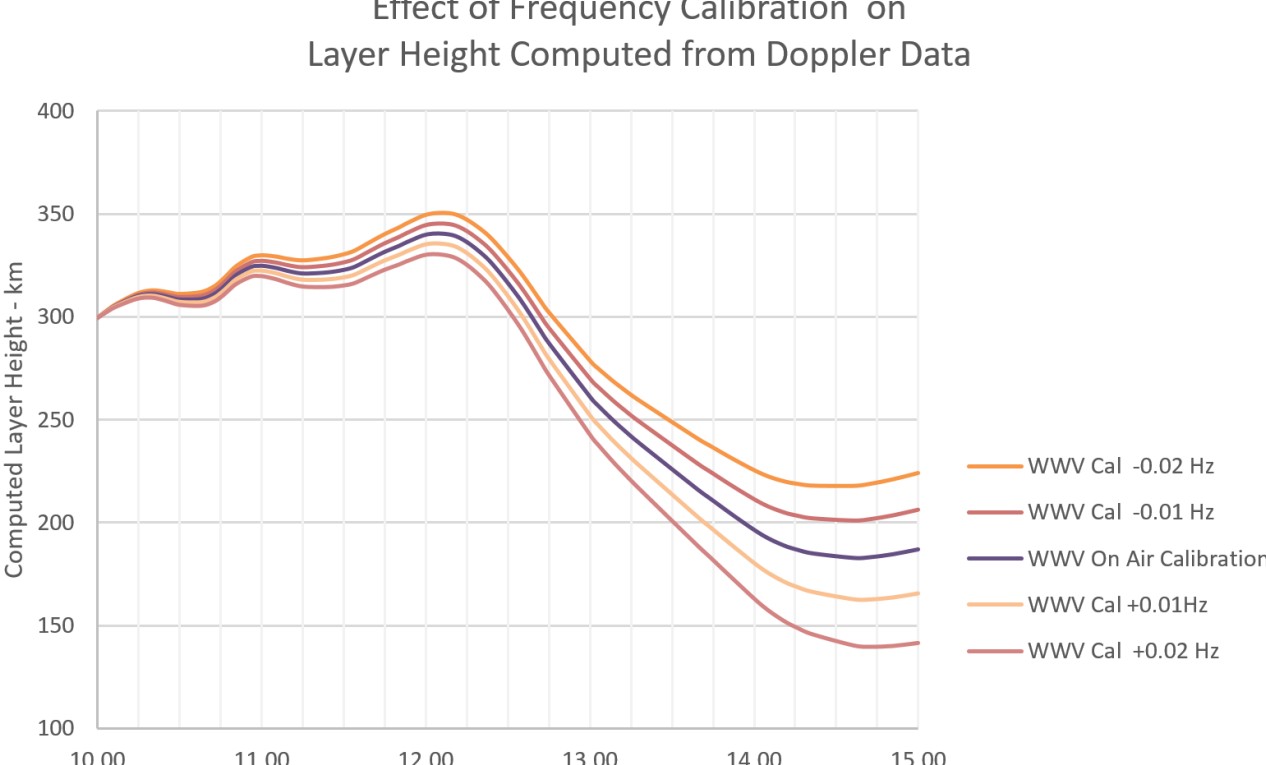

**Figure 16.** Effect of frequency calibration on layer height computed from Doppler data. Over time, a slight error in calibration can produce an error in layer height estimation on the order of tens of kilometers.

differentiating a smoothed version of ionosonde hmF2 data after converting layer height to path length for the different modes. Effective layer height change can be deduced by the inverse integration process on measured Doppler data and reconciliation with path length through the use of a geometric model. Several experiments suggest that the height changes responsible for Doppler shifts occur at the F layer. In contrast, the E layer shows relative height stability in the face of diurnal transition periods and eclipse passages. Signals believed to be refracted from the E layer show comparative frequency stability.

This study has presented data collected using relatively inexpensive tools widely available to hobbyists. The potential of this avenue for distributed measurement is well-established but not yet fully explored. Our study results, in particular the comparison with ionosonde data, suggest possibilities for targeted future campaigns. For example, Doppler studies can be conducted from receiving stations placed such that an ionosonde is situated at the approximate apogee of the path from WWV. Another prospect is to correlate data from multiple strategically placed receivers with data from multiple ionosondes, mapping the bottomside ionosphere over the continental US. The Amateur Radio Science Citizen Investigation (HamSCI) is also exploring modalities



to incorporate quantitative ionospheric measurements into the recreational activities of amateur radio operators, in order to improve sensing infrastructure while energizing the citizen science community.

Advancements in instrumentation and data collection will also help to achieve this goal. The timing measurements reported
here were obtained by manual measurements on oscilloscope recordings. Trace selection and interpretation were done by a human operator and therefore subject to bias. A more precise phase delay measurement may be possible through an algorithmic Fourier or related analysis of the composite waveform. An automated collection and data extraction method would give more complete and accurate data, and is under development by HamSCI. This software will be made available to citizen scientists running purpose-built Doppler measuring stations as part of the emerging Personal Space Weather Station network. The first
prototype of these Doppler stations is documented in Gibbons et al. (2022).



*Data availability.* The datasets and simulation results used herein are compliant with FAIR data standards and available per Table 1. The ionosonde data is drawn from the Lowell digisonde network, and can be accessed via https://giro.uml.edu/didbase/. Where applicable, datasets collected by amateur radio operators are identified by callsign.

*Author contributions.* Conceptualization of this work was led by Cerwin and Frissell. Data curation was performed by Collins. Formal
Analysis was performed by all authors. Funding acquisition was performed primarily by Frissell. Investigation was conducted by all authors, primarily Cerwin. Methodology was developed by all authors. Project administration was performed by Frissell. Resources were provided by all authors. Software was provided by Huba. Supervision was provided by Frissell. Validation was performed by all authors. Visualization was performed by Cerwin and Collins. Writing was led by Collins. Review and editing was led by Erickson.

*Competing interests.* The authors declare that they have no conflict of interest.

*Acknowledgements.* This work was undertaken by members of HamSCI (www.hamsci.org). We are indebted to the late Professor Steve Reyer WA9VNJ for his data collection during the 2017 eclipse. Support for this work was provided by NSF Grants AGS-2002278, AGS-1932997, and AGS-1932972. The contribution by JDH was supported by NSF grant AGS-1931415. We thank the Lowell GIRO Data Center (http://spase.info/SMWG/Observatory/GIRO) and the American taxpayer for supporting the ionosonde used herein. Eclipse paths were provided by Dr. Shunrong Zhang. The results published in this paper were obtained using the HF propagation toolbox, PHaRLAP, created by
Dr Manuel Cervera, Defence Science and Technology Group, Australia (manuel.cervera@dsto.defence.gov.au). This toolbox is available by request from its author. SAMI3 outputs are available by contacting J. Huba at huba@nrl.navy.mil. We acknowledge the use of the Free Open Source Software projects used in this analysis: Ubuntu Linux, python, matplotlib, NumPy, SciPy, pandas, and others. This work made use of the High Performance Computing Resource in the Core Facility for Advanced Research Computing at Case Western Reserve University. The authors thank David Casente, Joanna Elia and Marius Mereckis for the use of their processing code to identify WWV and WWVH. We thank
Carl Luetzelschwab for providing Proplab-Pro raytraces during the conception of this work, and acknowledge Solar Terrestrial Dispatch, the producers of that software. We also thank NIST and the staff at WWV and WWVH, without whom this work would not be possible.



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
