# Peer review of "Methods for Estimation of Ionospheric Layer Height Characteristics from Doppler Frequency and Time of Flight Measurements on HF Skywave Signals"

_EGUsphere, 2022_

## Referee Comment (RC1)

**Title:** Methods for Estimation of Ionospheric Layer Height Characteristics from Doppler Frequency and Time of Flight Measurements on HF Skywave Signals, (Collins et al., forthcoming)

**Summary:**

The paper demonstrated a methodology to estimate ionospheric virtual layer height characteristics using Doppler measurements from frequency locked time standard stations in conjunction with ionosonde measurements and ray-tracing models. The study showed observations are consistent with a model in which mode splitting originates from different path length velocities associated with single and multiple hop modes as the virtual layer height changes. The study took the support of complementary processes of 1) calculating Doppler shifts from virtual layer height changes and virtual layer height changes from Doppler shifts, and 2) the analysis of intermittent low-Doppler shift modes including correlation with ionosonde observations to help identify multihop propagation modes to justify authors claim.

**Overall remarks:**

The work is significant and covers a timely topic that aids the analysis to understand various space weather effects on the ionosphere using path length estimates from distributed HF stations when integrated with other ionospheric measurements. The major drawback of the manuscript is its organization and readability. The figures are not described in the text properly, and neither they are justified. The chronology of the figure number is awkward, Figure 11-13 is mentioned before Figure 10 in the text. Also, not all the figures are labeled and some of the figures are not of publication quality. Finally, the reviewer was not able to find a proper justification or necessity of the methodology in the manuscript, in the introduction, discussion, and conclusion sections.

I recommend addressing the following points before proceeding with the publication.

1. Line 5-6: 'mode splitting … layer height changes' – not clear, please rewrite the sentence.
2. Line 8-9: '2) the analysis … propagation modes.' – not clear, please rewrite the sentence.
3. Line 19: '… condition change, time-dependent changes…' – can it be related to spatial change also?
4. Line 23: '… studies of ionospheric conditions **()**.' – provide at least one citation.
5. Line 39: '… and layer width …' – change to 'thickness'.
6. Line 41: '… in order to stabilize the solution.' – What solution? Why does the author need to stabilize the solution?
7. Line 42-43: 'In this study, … to ionospheric height.' – complex idea, please break down into two sentences.
8. Line 79: 'The composition …' – not sure what composition.
9. Line 81: '… distributed density …' – not clear.
10. Line 89: '… ionospheric electron content **()**.' – provide citations.
11. Line 101: '… a suitable ….' How does the author know what is suitable?
12. Figure 2 and the parameters used in there are not described in detail in the text.
13. An additional figure is required to describe the working principle of the method that is described in Figure 3.

14. The reviewer is not able to understand step 3 of the methodology.
15. Line 122: Equation 3 should be replaced by numerical integration, such as the Trapezoidal rule.
16. The reviewer is not sure about step 5 of the methodology.
17. It is better to describe HF RT (section 2.3) before section 2.2, as it is referenced in Figure 3 and methodology.
18. Most of the figures (7-15) lack motivation (Why is it needed in the paper? What question does it answer?), location in the manuscript (text mentioned in Figures 11-13 before 10), and labels (x, and y scales), which make them very difficult to interpret and justify their needs.
19. Line 247-249: '… give comparable …', not sure authors presented this comparison anywhere in the manuscript. Please provide additional details of this comparison.
20. Provide a clear objective and findings of the study in the introduction and conclusion section of the paper.
21. The discussion section should compare the results in the context of previous research, but here authors discussed the figures and observations in the figures.

Considering all these comments together, I would strongly recommend restructuring the paper by rearranging the sections (sub-sections), figures, and discussion sections.

---

## Author Comment (AC1)

We thank the reviewers for their attention and helpful comments on this manuscript.

We have made substantial revisions, including restructuring the sections and recreating or consolidating figures. In particular, the dataset from 1 October 2019, which was taken without a GPSDO, has been replaced with a dataset from 11 May 2022. The eclipse discussion has also been removed in order to improve the focus of the paper.

Our responses to Reviewer 2 are included inline below.
* * *
**https://doi.org/10.5194/egusphere-2022-327-RC2**

**The paper presents three events and analysis to understand Doppler shifts observed using a few different methods, including by amateur radio operators.  One event corresponded to the eclipse of 2017, and two other events corresponded to Doppler shifts associated with the terminator.**

**While the work that forms the investigation has merit and presents some novel observations, particularly with respect to the fact that the equipment used is amateur radio equipment, the new method being presented in the paper is unclear.  I could not determine what was really the new science or the new technique/methodology presented in the paper.  From the science perspective, there have been previous investigations of Doppler shifts during eclipses, at minimum, but those were not compared with the current results.  There are two new aspects from this investigation which are important and should be emphasized.  First, the observations were made with amateur radio equipment, so presumably some of these calibrations could be described and it is possible for others to replicate these results.  Second, using ticks with WWV forms a new technique.**

**I recommend that this paper focuses a little bit more or adds some text to tie these events together into a more cohesive story.**

The paper has been restructured to focus on the two methodologies applied: the estimation of layer height from Doppler shift and the measurement of time of flight via WWV second ticks.

**I have the following major comments:**

1. **The paper is disconnected in terms of connecting these three experiments together. Frankly, they seem like three discrete experiments that are loosely connected, although the second and third experiment seem different than the eclipse experiment.  There needs to be more effort put toward having a cohesive line of logic for the reader.  One suggestion might be to simply not**

**talk about the eclipse results and focus instead on the second and third experiment since these seem like a more cohesive story.**

We have removed the eclipse results and significantly restructured the flow of the paper in keeping with this recommendation.

2. **The paper seems to suggest estimating the virtual height using Doppler observations. From a pure radar signal processing perspective this doesn't make sense since a narrow band CW signal has an infinite range resolution.**
That is correct. We are measuring change in virtual height, not the virtual height itself.

   **If the intent is instead to correlate Doppler variations with virtual height variations, more modeling effort or theory is required to demonstrate that this can be done feasibly. In particular in Section 2, there should be some additional figures that demonstrate this methodology more clearly and what sort of results are being found. So for example, do you see systematic trends in the virtual height relative to Doppler? Those should be explained clearly and including figures. If this is the crux of the new technique, there should be more justification demonstrating that this technique works.**
Three figures from the original manuscript (5, 8 and 9) have been replaced with analogous subfigures in Figure 7, in order to more coherently present the steps of the integrated Doppler methodology by example.

3. **You should quantify the errorbars on the virtual height estimation. This also may illustrate my point that if you use Doppler alone, you will end up with enormous errorbars. If that is the case, what conclusion can you draw about the virtual height? Regardless, the errorbars would help in terms of the quality of the investigation.**
Error bars have been added to the spectrogram analysis, and a more extensive discussion of error has been added.

4. **The flow charge in Figure 3 and the enumerated list in Section 2 are confusing. I could not understand how this method/technique actually worked. This needs to be clarified and elaborated, perhaps with an example. What is the basis for equations 2 and equation 3? This was not explained.**
The flowchart has been rewritten and more text has been added to contextualize the steps and add clarity. The corresponding plots have been organized into a set of ordered subfigures, which are referenced within the flowchart, in order to improve the flow of this section.

5. **Near line 185, you have a sentence that states "Figures 8 and 9 show changes in the path velocity and length calculated…" this is a single sentence that describes two figures. What are the key take away points you want the reader to see in each of these figures? The single sentence is insufficient in the description. Also how were these quantities calculated in Figure 8? There are a lot of arrows and other things happening in the figure without a clear description in the text.**

   The purpose of these figures was to illustrate intermediate steps in the Doppler layer height method. The figures have been replotted and consolidated as subfigures in Figure 7. The python code used to produce the new versions is available in the Open Research section.

6. **Figure 5 shows some eclipse data for a control day and the day of the eclipse. During 1400-1600 UT, the Doppler measurements appear to have similar magnitudes on the control day versus the day of the eclipse? Why is that the case? I think this would benefit from using some temporal smoothing – like a running mean or median. The trends should be clearer.**

   The eclipse section has been removed.

7. **In Figure 9 and 10, I am confused when the Ionosonde data is used relative to your estimates of the virtual height? It does seem cyclic to me to use ionosonde data for a virtual height and then is your algorithm modifying the virtual height to match the doppler observations? Please clarify this in the text.**

   A clarification has been added to Step 6. "[T]he Boulder ionosonde is used as a validation tool: the frequency profiles calculated from the Doppler shift and a single initial measurement are compared to the profiles measured subsequently by the ionosonde."

8. **Figure 12 should have error estimates associated with the data points.**

   For the time of flight measurements (plotted in Figure 12 in the original manuscript), the measurement uncertainty is considered negligible. This is discussed in greater detail in Section 3.5.2 of the revised manuscript.

---

## Author Comment (AC2)

We thank the reviewers for their attention and helpful comments on this manuscript.

We have made substantial revisions, including restructuring the sections and recreating or consolidating figures. In particular, the dataset from 1 October 2019, which was taken without a GPSDO, has been replaced with a dataset from 11 May 2022. The eclipse discussion has also been removed in order to improve the focus of the paper.

Our responses to Reviewer 1 are included inline below.

─────────────────────────────────────────────────────────────

**https://doi.org/10.5194/egusphere-2022-327-RC1**

**Title: Methods for Estimation of Ionospheric Layer Height Characteristics from Doppler Frequency and Time of Flight Measurements on HF Skywave Signals, (Collins et al., forthcoming)**

**Summary:**
**The paper demonstrated a methodology to estimate ionospheric virtual layer height characteristics using Doppler measurements from frequency locked time standard stations in conjunction with ionosonde measurements and ray-tracing models. The study showed observations are consistent with a model in which mode splitting originates from different path length velocities associated with single and multiple hop modes as the virtual layer height changes. The study took the support of complementary processes of 1) calculating Doppler shifts from virtual layer height changes and virtual layer height changes from Doppler shifts, and 2) the analysis of intermittent low-Doppler shift modes including correlation with ionosonde observations to help identify multihop propagation modes to justify authors claim.**

**Overall remarks:**
**The work is significant and covers a timely topic that aids the analysis to understand various space weather effects on the ionosphere using path length estimates from distributed HF stations when integrated with other ionospheric measurements. The major drawback of the manuscript is its organization and readability.**

We have significantly restructured the revised manuscript in order to improve its readability.

**The figures are not described in the text properly, and neither they are justified. The chronology of the figure number is awkward, Figure 11-13 is mentioned before Figure 10 in the text. Also, not all the figures are labeled and some of the figures are not of publication quality.**

The figure order has been reworked, and figures related to each of the methodologies (Doppler path length integration, time of flight measurement) have been grouped into subfigures in order to demonstrate the methodologies more clearly.

**Finally, the reviewer was not able to find a proper justification or necessity of the methodology in the manuscript, in the introduction, discussion, and conclusion sections.**

A motivation section has been added to more clearly describe the phenomenon of interest (mode splitting) which prompted the investigations described in the paper. The introduction and conclusion sections have also been revised to provide more extensive discussion of these methods' utility, particularly with regard to analysis of data from the Personal Space Weather Station network.

From the revised introduction section:

*"Recently, interest has grown in developing crowd sourced, distributed networks of HF receiver stations for geospace environmental monitoring. These meta-instruments (i.e., networks of small instruments effectively operating as a single large instrument), operated by radio amateurs and shortwave listeners, have the potential to greatly improve the density of instrumentation for bottomside ionospheric sampling. Recent studies by several authors (Frissell et al., 2018, 2019, 2022; Perry et al., 2018; Collins et al., 2022) have demonstrated the scientific validity of qualitative HF observations in these types of networks, using existing amateur radio equipment. Subsequent processing of these observations for scientific studies of ionospheric conditions and variability requires further analysis to extract information from the direct Doppler observations. This is particularly true given the non-unique nature of the measurements since changes in ionospheric parameters such as peak electron density, layer height, and layer thickness can produce identical Doppler shifts (Lynn, 2009; Chilcote et al., 2015). Analysis must therefore use reasonable ionospheric models, prior information, or other equivalent techniques in order to differentiate among the ionospheric parameter state changes causing the observed variations.*

**I recommend addressing the following points before proceeding with the publication.**
1. **Line 5-6: 'mode splitting ... layer height changes' – not clear, please rewrite the sentence.** An additional expository sentence has been added: "This study is motivated by the phenomenon of `mode splitting,' the observation of multiple simultaneous propagation paths, or `modes.'"
2. **Line 8-9: '2) the analysis ... propagation modes.' – not clear, please rewrite the sentence.** An additional expository sentence has been added, as above.
3. **Line 19: '... condition change, time-dependent changes...' – can it be related to spatial change also?** This sentence has been clarified to reflect that a stationary receiver is assumed.
4. **Line 23: '... studies of ionospheric conditions ().' – provide at least one citation.** A citation has been added: (Breit and Tuve, 1925).
5. **Line 39: '... and layer width ...' – change to 'thickness'.**
   This change has been made.
6. **Line 41: '... in order to stabilize the solution.' – What solution? Why does the author need to stabilize the solution?**

This sentence has been reworded for clarity: "Analysis must therefore use reasonable ionospheric models, prior information, or other equivalent techniques in order to differentiate among the ionospheric parameter state changes causing the observed variations."

7. **Line 42-43: 'In this study, ... to ionospheric height.' – complex idea, please break down into two sentences.**
   This change has been made: "In this study, we target ionospheric refraction height information through analysis of HF Doppler receiver observations. Our quantitative approach relates Doppler shift to change in ionospheric height."

8. **Line 79: 'The composition ...' – not sure what composition.**
   For clarity, this has been revised to "the Fourier components *in the received signal."*

9. **Line 81: '... distributed density ...' – not clear.**
   This sentence has been removed.

10. **Line 89: '... ionospheric electron content ().' – provide citations.**
    A citation has been added: (Chakraborty et al., 2022).

11. **Line 101: '... a suitable ....' How does the author know what is suitable?**
    More expository text has been added here: "...using a suitable time increment between data points, such that the shape of the trace is accurately represented. In the case of the 1-hop F layer trace in the manually digitized data (...), for example, 84 points were used over a data record spanning 2.3 hours, for an average rate of about 1 data point every 1.7 minutes."

12. **Figure 2 and the parameters used in there are not described in detail in the text.**
    The illustration of the geometric Martyn height equivalent model (Figure 2 in the original manuscript) is discussed in greater detail in the Doppler methodology section in the revised manuscript, particularly in Step 5.

13. **An additional figure is required to describe the working principle of the method that is described in Figure 3.**
    Three figures from the original manuscript (5, 8 and 9) have been replaced with analogous subfigures in Figure 7, in order to more coherently present the steps of the integrated Doppler methodology by example.
    The flowchart that constituted Figure 3 in the original manuscript has been revised to connect it more closely to the corresponding subfigures and steps of the methodology.

14. **The reviewer is not able to understand step 3 of the methodology.**
    In the revised version of the manuscript, this step has been incorporated into the trapezoidal integration step. The notation is more unified and should be easier to follow.

15. **Line 122: Equation 3 should be replaced by numerical integration, such as the Trapezoidal rule.**
    This change has been implemented.

16. **The reviewer is not sure about step 5 of the methodology.**
    This step describes the selection of an initial reference layer height estimate from an ionosonde or other initial source. (Due to the revision of earlier steps, it is Step 4 in the revised manuscript.) We have added additional exposition in Step 4 and Step 7 to describe the role of the ionosonde data as a validation tool.

17. **It is better to describe HF RT (section 2.3) before section 2.2, as it is referenced in Figure 3 and methodology.**
   This subsection has been moved accordingly within the Methodology section.
18. **Most of the figures (7-15) lack motivation (Why is it needed in the paper? What question does it answer?), location in the manuscript (text mentioned in Figures 11-13 before 10), and labels (x, and y scales), which make them very difficult to interpret and justify their needs.**
   Several of these figures have been reordered and consolidated in the revised manuscript. Some have also been replotted in order to increase consistency and clarity.
19. **Line 247-249: '... give comparable ...', not sure authors presented this comparison anywhere in the manuscript. Please provide additional details of this comparison.**
   This comparison is presented in a new figure (Fig. 3) which shows a forward model generating mode splitting effects from smoothed ionosonde data.
20. **Provide a clear objective and findings of the study in the introduction and conclusion section of the paper.**
   We have added a motivation section which contextualizes our investigation of mode splitting, and a clear list of key points has been added to the Conclusions section.
21. **The discussion section should compare the results in the context of previous research, but here authors discussed the figures and observations in the figures.**
   We have rewritten the discussion section to place the results in a research context. Mode splitting, which was the focus of the discussion section in the original manuscript, is instead covered in the Motivation section.

**Considering all these comments together, I would strongly recommend restructuring the paper by rearranging the sections (sub-sections), figures, and discussion sections.**
 These sections have been restructured.